# Hierarchical Classification via Diffusion on Manifolds

## Abstract

Hierarchical classification, the problem of classifying images according to a predefined hierarchical taxonomy, has practical significance owing to the principle of "making better mistakes", i.e., better to predict correct coarse labels than incorrect fine labels. Yet, it is insufficiently studied in literature, presumably because simply finetuning a pretrained deep neural network using the cross-entropy loss on leaf classes already leads to good performance w.r.t not only the popular top-1 accuracy but also hierarchical metrics. Despite the empirical effectiveness of finetuning pretrained models, we argue that hierarchical classification could be better addressed by explicitly regularizing finetuning w.r.t the predefined hierarchical taxonomy. Intuitively, with a pretrained model, data lies in hierarchical manifolds in the feature space. Hence, we propose a hierarchical multi-modal contrastive finetuning method to leverage taxonomic hierarchy to finetune a pretrained model for better hierarchical classification. Moreover, the hierarchical manifolds motivate a graph diffusion-based method to adjust posteriors at hierarchical levels altogether in inference. This distinguishes our method from the existing ones, including top-down approaches (using coarse-class predictions to adjust fine-class predictions) and bottom-up approaches (processing fine-class predictions towards coarse-label predictions). We validate our method on two large-scale datasets, iNat18 and iNat21. Extensive experiments demonstrate that our method significantly outperforms prior arts w.r.t both top-1 accuracy and established hierarchical metrics, thanks to our new multi-modal hierarchical contrastive finetuning and graph diffusion-based inference.

## 1 Introduction

Hierarchical classification has long been a pivotal and challenging problem in the literature Naumoff (2011); Deng et al. (2012); Zhu & Bain (2017); Bertinetto et al. (2020). It aims to categorize images w.r.t a given hierarchical taxonomy, adhering to the principle of "making better mistakes", which essentially favors correct coarse-class predictions over inaccurate fine-class predictions Deng et al. (2012); Wu et al. (2020).

Methods of hierarchical classification improve either training or inference. Existing inference methods can be divided into two types: top-down Redmon & Farhadi (2017) and bottom-up Valmadre (2022). Top-down methods adjust the posterior for predicting a specific class by using its parent/ancestor posterior probabilities. They often underperform bottom-up methods Redmon & Farhadi (2017); Bertinetto et al. (2020), which prioritize predicting the leaf-classes and subsequently calculate posteriors for the parent/ancestor classes. Valmadre (2022) attributes the underperformance of top-down methods to the high diversity within coarse-level categories, soliciting effective training methods. Perhaps surprisingly, although these sophisticated hierarchical classification methods show promising results in certain metrics, they do not consistently rival the simplistic flat-softmax baseline Valmadre (2022), which learns a softmax classifier on the leaf classes only. The status quo leads to a natural question: *Is it still helpful to make predictions for hierarchical classes other than the leaf classes for better hierarchical classification?* That said, it is still an open question how to effectively exploit hierarchical taxonomy to improve training and inference for hierarchical classification.

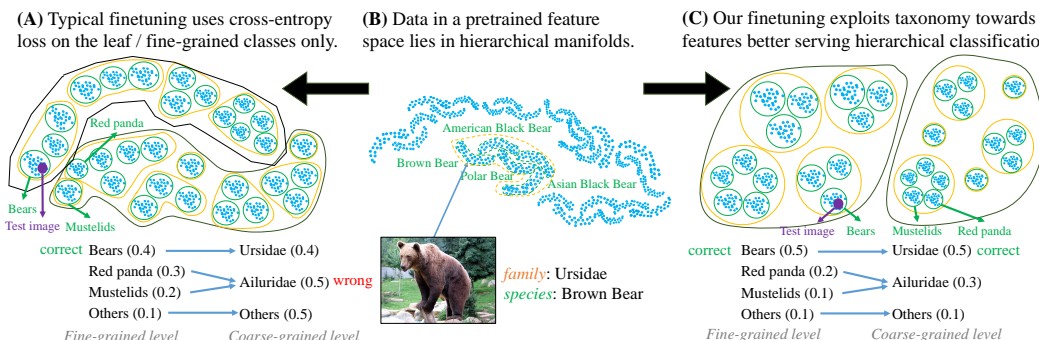

Figure 1: To solve a downstream task of classification, a *de facto* practice is finetuning a pretrained model using the cross-entropy loss on leaf classes (e.g., Brown Bear at the species level). **(A)**: This yields features that help leaf-class classification but fail to model their hierarchical relationships w.r.t the predefined taxonomy (e.g., Ursidae at the family level). That said, learning with species labels only does not necessarily help hierarchical classification. Nevertheless, such features are better than the "raw features" of the pretrained model, which provides a feature space **(B)** where data hypothetically lie in hierarchical manifolds w.r.t the taxonomy. **(C)**: Differently, we propose to finetune the pretrained model by *explicitly* exploiting the hierarchical taxonomy towards features that can better serve the task of hierarchical classification (Figure 2), e.g., finetuned features well reflect the defined hierarchical taxonomy.

We first propose to collectively adjust posteriors at multiple hierarchical levels towards the final results of hierarchical classification. To this end, we present a set of graph diffusion-based methods for inference (Section 3.2), inspired by the literature of information retrieval Page et al. (1998); Iscen et al. (2017); An et al. (2021) which shows that diffusion is adept at mapping manifolds. This distinguishes our methods from existing top-down and bottom-up inference approaches that linearly interpret hierarchical classification. Our methods treat the hierarchical taxonomy as a graph, enabling probability distribution in the taxonomy. To the best of our knowledge, our work makes the first attempt to apply graph diffusion to hierarchical classification. Extensive experiments demonstrate that our graph diffusion-based inference methods, along with HMCF, achieve state-of-the-art performance and resoundingly outperform prior arts (Section 4.3).

Furthermore, we propose a Hierarchical Multi-Modal Contrastive Fine-Tuning (HMCF) strategy (Section 3.3) to leverage the hierarchical taxonomy for learning more representative features that align with the taxonomy and enhance hierarchical classification. While prior research has validated the effectiveness of vision-language models (VLMs) in standard image classification Xiao et al. (2022); Jin et al. (2021), this study investigates their utility in hierarchical classification by quantifying performance improvements and evaluating their ability to tackle the manifold challenge.

To summarize, we make three major contributions.

1. We revisit the problem of hierarchical classification from the perspective of manifold learning, offering new insights in the contemporary deep learning land.

2. We introduce a novel graph diffusion-based inference method to exploit posteriors at multiple levels towards the final prediction.

3. We present the hierarchical multi-modal contrastive finetuning strategy for finetuning a VLM to better solve the problem of hierarchical classification.

## 2    RELATED WORK

**Hierarchical classification** is of practical significance owing to the goal of predicting correct coarse-level labels if predicting fine-level ones is too difficult. Datasets like ImageNet Russakovsky et al. (2015) and WordNet Miller (1995) have long emphasized taxonomy, while newer ones like iNat18 Van Horn et al. (2018) and iNat21 Van Horn et al. (2021) offer finer-grained labels. Research in this domain has shown significant progress, with fundamental studies like "Hedging Your Bet" Deng

et al. (2012) and contemporary deep learning approaches employing flat softmax, softmargin, and descendant softmax training losses Valmadre (2022), along with bottom-up Valmadre (2022) and top-down Redmon & Farhadi (2017) inferences. Its practical applications are evident in areas like long-tailed 3D detection for autonomous driving Peri et al. (2023), emphasizing specific metrics, methods, and joint training. Despite extensive research, recent findings suggest that advanced training and inference methods do not consistently surpass the flat softmax baseline Valmadre (2022). We present innovative techniques that harness hierarchical data more efficiently during both training and inference.

**Fine-grained visual categorization** is a task bridging coarse-level classification and instance-level classification, presents both significant value and substantial challenges Akata et al. (2015); Yang et al. (2018). In cases where predicting classe at the fine-grained level is erroneous, users often prefer an accurate coarse-level result, highlighting the importance of hierarchical classification within the fine-grained classification area Deng et al. (2012). This paper contributes to this aspect, pushing forward the understanding and application of hierarchical fine-grained categorization in the context of long-tail distributions.

**Visual Language Models (VLMs)** has gained significant attention in the research community, particularly following the introduction of OpenAI's CLIP Radford et al. (2021) and Google's ALIGN Jia et al. (2021). These models are extensively employed in various tasks, including visual question answering Antol et al. (2015), language-guided image generation Jiang et al. (2021), and vision-language navigation Zhu et al. (2020). Despite their widespread use, there is a lack of application of VLMs in hierarchical classification problems to date. This paper posits that taxonomies in hierarchical classification encompass not only a hierarchical arrangement of concepts (such as species, genus, order, family, etc.) but also descriptive texts or names associated with these concepts. We investigate the application of VLMs in hierarchical classification for the first time, exploring their potential effectiveness in this novel context.

**Graph diffusion** is an advanced methodology adept at faithfully delineating the manifold within a data distribution by leveraging the inter-connectedness inherent in a Markov chain Zhou et al. (2003a;b). The renowned variation of this method PageRank Page et al. (1998) has achieved considerable success in various business endeavors. Moreover, it has been extensively employed in the area of image retrieval Iscen et al. (2017); An et al. (2021), an application of instance-level classification. However, its potential in broader classifications, such as fine-grained and hierarchical categorizations, has not been extensively explored. In this paper, we explore graph diffusion for hierarchical classification, motivated by current practice of adjusting posteriors of all categories using the given hierarchical taxonomy.

## 3 METHODS

**Notations and problem definition**. Let $Y$ denote the set of all the categories within the taxonomy tree. Every node in $Y$ is a category. $C(y)$ and $A(y)$ index the children and ancestors of category $y \in Y$, respectively. $B$ is the set of bottom nodes (i.e., leaf nodes), and $B(y)$ denotes the leaf nodes which are the descendants of $y$. We call $y \in (Y - B)$ the intermediate nodes. For an image $x$, the problem of hierarchical classification requires a classifier to predict any category within $Y$, not being confined to only the leaf nodes.

### 3.1 HIERARCHICAL MANIFOLDS

**Status quo.** To predict the intermediate categories for an input image, existing methods can be divided into two approaches: top-down Redmon & Farhadi (2017); Jain et al. (2023) and bottom-up Valmadre (2022); Wu et al. (2020). Top-down methods adjust the posterior to predict a specific category by using its parent/ancestor posterior probabilities. Bottom-up methods Redmon & Farhadi (2017); Bertinetto et al. (2020) directly predict the leaf categories and subsequently calculate posteriors for the parent/ancestor categories.

Two key observations emerge from this distinction. First, despite the elegance of top-down methods in utilizing parent probabilities, they often underperform when compared to bottom-up methods Valmadre (2022), which do not rely on explicit neural network predictions for intermediate category probabilities. Second, there are cases where bottom-up methods, despite successfully predicting a

leaf-level category, surprisingly predict incorrect mid-level categories. These observations lead to an important question:*Can the predictions across different levels in the category hierarchy mutually reinforce each other to improve overall accuracy?*

**Hypothesis.** We assume that the reason for the observations we mentioned above is the existence of the hierarchical manifolds; examples from the same category in the feature space lie not only in the manifold but also hierarchically in manifolds w.r.t different levels of labels, as illustrated in Figure 1. In plain language, parent manifolds (corresponding to the coarse level of labels) envelop child manifolds (corresponding to the fine level of labels).

Hierarchical manifolds introduce challenges that prevent predictions at different levels from effectively supporting each other. For instance, as illustrated in Figure 1, even if the top-1 prediction at the leaf level (e.g., "bear") is correct, the model might still incorrectly predict a mid-level category, such as "Ailuridae," due to the hierarchical manifolds. To fully leverage intermediate-level probabilities, it is crucial to account for the hierarchical manifold problem during both training and inference. In the following sections, we first present our novel inference method, followed by a detailed description of our training approach.

## 3.2 GRAPH DIFFUSION-BASED INFERENCE

In this work, we introduce an advanced inference method to tackle the hierarchical manifold. Different from the top-down inference Redmon & Farhadi (2017) that directly calculates a child's probability conditioned on its parent's probabilities - for instance, $P(\text{Norfolk terrier}) = P(\text{Norfolk terrier}|\text{terrier})P(\text{terrier})$ - our approach introduces a novel graph diffusion strategy. This strategy adjusts the probabilities of each nodes based on the predictions of the entire graph and the relationships of categories; we utilize graph diffusion to establish a stable distribution of scores throughout the taxonomy tree.

We first frame hierarchical classification as a ranking problem, where nodes in the taxonomy tree are ranked for each image. For example, given an image, we rank all nodes (e.g., 14,036 in iNat18 Van Horn et al. (2018)) so that the highest-ranked nodes correspond to the correct labels. While softmax is applied separately at each level, nodes from all levels can still be ranked together before softmax is applied. Unlike traditional top-down inference, this method does not require the parent node's probability to equal the sum of its children's probabilities, and loosening this condition does not negatively affect hierarchical classification. This approach enables the effective use of graph diffusion in later stages.

We apply graph diffusion as a post-processing step to refine the ranking results. This approach is motivated by the same principle as PageRank Page et al. (1998), where nodes connected to important nodes are also considered important. This method offers a distinct advantage over traditional top-down and bottom-up inference by addressing the manifold problem. When a node is misclassified, diffusion can leverage predictions from nodes across all levels to correct the error. For instance, if the model initially misclassifies a Chihuahua as a Sphynx cat, graph diffusion can transfer relatively high scores from related categories, such as terrier or labrador, back to Chihuahua, ultimately refining the prediction and correctly identifying the image as a Chihuahua. Below, we provide a detailed description of our graph diffusion-based method.

**Method details.** Our method diffuses prediction scores among categories defined by a taxonomy. Given a total of $n$ categories (including both leaf and intermediate ones) in the predefined taxonomy, we use a connection matrix $W \in R^{n \times n}$ to describe the relationships between categories. Specifically, $w_{ij} = 1$ if category $i$ and $j$ have a parent-children relation in the taxonomy; otherwise $w_{ij} = 0$. We assume undirected graph given a taxonomy, so the connection matrix is symmetric, i.e., $W = W^T$. The self-similarity is set to 0, i.e., $\text{diag}(W) = \mathbf{0}$. We explore more options for the connection matrix later. Importantly, following the literature Page et al. (1998); Iscen et al. (2017), we normalize the connection matrix as below:

$$\bar{W} = D^{-1/2}WD^{-1/2}, \quad D = \text{diag}(W\mathbf{1}). \tag{1}$$

Let $f^0 \in R^n$ be the vector of prediction scores for the $n$ categories. Our goal is to adjust $f^0$ towards refined ones (denoted by $f^\star$) by considering all the scores and the relationships among categories. Specifically, we propose to diffuse the scores over the graph specified by the connection matrix $\bar{W}$.

$\mathbf{1}$ is a vector whose values are 1 and $W\mathbf{1}$ is a normalized Laplacian matrix. The diffusion process iteratively updates the category scores:

$$f^{t+1} = \alpha\bar{W}f^t + (1-\alpha)f^0,\tag{2}$$

where $\alpha \in (0,1)$ is a hyperparameter. This process is a "random walk" algorithm Page et al. (1998). Intuitively, in an iteration, each category spreads its prediction score to its neighbor categories with a probability $\alpha$ and takes the initial prediction with a probability $1-\alpha$.

**Convergence analysis**. The iterative process of graph diffusion above is assured to converge towards a stationary distribution Zhou et al. (2003b). We provide a straightforward proof here. By recursively iterating $f^1 = \alpha\bar{W}f^0 + (1-\alpha)f^0$ into subsequent iterations $f^t$, we derive:

$$f^t = (\alpha\bar{W})^t f^0 + (1-\alpha)\sum_{i=0}^{t}(\alpha\bar{W})^i f^0.\tag{3}$$

As $t$ approaches infinity, the term $(\alpha\bar{W})^t$ approaches zero because $\alpha \in (0,1)$ and $\bar{w}_{ij} \in [0,1]$. The summation term converges to $(I - \alpha\bar{W})^{-1}$, where $I$ denotes an identity matrix of size $n$; the summation term is its power series representation. Thus, the eventual stationary distribution is:

$$f^* = (1-\alpha)(I - \alpha\bar{W})^{-1}f^0.\tag{4}$$

**Differentiable diffusion**. Equation 4 shows that the graph diffusion converges to a closed form. Intriguingly, this represents a linear transformation (i.e., the transform mapping given by $(1-\alpha)(I - \alpha W)^{-1}$) of the initial scores $f^0$. Hence, it is intuitive to replace the connection matrix $W$, which is constructed based on the predefined taxonomy, to another which can be learned to better serve hierarchical classification. Therefore, we explore learning such a linear transform directly from data. In practice, we learn such a transform matrix, taking as input the initial prediction scores $f_0$, by minimizing the cross-entropy loss over training data. We call this learning-based transform mapping *differentiable diffusion*.

**Remark** We note two advantages of our diffusion approach over existing top-down and bottom-up methods:

1. *Leveraging predictions of all categories.* Unlike many existing methods that post-hoc derive scores using parent-child relationships, ours exploits the entire graph structure defined by the taxonomy, allowing adjusting scores by considering all categories at once.

2. *Handling data manifolds.* Graph diffusion is well-known for handling data manifolds Page et al. (1998); Iscen et al. (2017). Hence, using diffusion to tackle the hierarchical manifolds intuitively better serves hierarchical classification than existing methods, which do not yet exploit data manifolds.

### 3.3 LEARNING WITH HIERARCHICAL TAXONOMY

In addition to inference, training plays a crucial role in addressing hierarchical manifolds. As illustrated in Figure 1-C, explicitly leveraging the hierarchical taxonomy during training can lead to features that better support hierarchical classification. However, many existing hierarchical classification methods generally boil down to the strategy of learning with leaf-level labels only Valmadre (2022). For instance, given a training image, the flat softmax method employs bottom-up inference for predicting score $q$ of interior node $y$ for the input image $I$ via the formula below Valmadre (2022):

$$q_y(I;\theta) = \begin{cases} [\text{softmax}_B(I;\theta)]_y & \text{if } y \in B \\ \sum_{v\in C(y)} q_v(I;\theta) & \text{if } y \notin B, \end{cases}\tag{5}$$

where $\theta$ is the model parameters. The negative log-likelihood concerning the high-level nodes is *reduced to the leaf nodes* as

$$\ell(y; I, \theta) = -\log q_y(I;\theta) = -\log\left(\sum_{y_i \in B(y)} \exp s_i\right) + \log\left(\sum_{y_i \in B} \exp s_i\right),\tag{6}$$

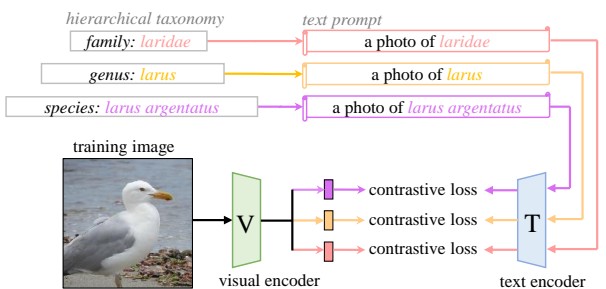

Figure 2: The proposed hierarchical multi-modal Contrastive Finetuning (HMCF) exploits hierarchical taxonomy to adapt a pretrained visual encoder to the downstream task of hierarchical classification. It sums contrastive losses between a training image and its taxonomic names at multiple levels.

where $s_i$ is the prediction score for category $y_i$. Advanced losses, such as soft-margin and descendant softmax Valmadre (2022), also focus on the leaf level, without effectively leveraging hierarchical labels in learning, hence may achieve suboptimal performance of hierarchical classification.

In this work, we utilize hierarchical textual descriptions to explicitly leverage the hierarchical taxonomy. We introduce *hierarchical multi-modal contrastive finetuning* (HMCF) to finetune a VLM for hierarchical classification (cf. Figure 2). HMCF exploits contrastive losses Goyal et al. (2023) built at $L$ hierarchical levels:

$$\mathcal{L} = \sum_{l=1}^{L} \Big( \sum_{i=1}^{N} - \log \frac{\exp(\mathcal{V}^l(I_i) \cdot \mathcal{T}(t_i^l))}{\sum_{j=1}^{N} \exp(\mathcal{V}^l(I_i) \cdot \mathcal{T}(t_j^l))} +$$
$$\sum_{i=1}^{N} - \log \frac{\exp(\mathcal{V}^l(I_i) \cdot \mathcal{T}(t_i^l))}{\sum_{j=1}^{N} \exp(\mathcal{V}^l(I_j) \cdot \mathcal{T}(t_i^l))} \Big),$$

where $N$ is the number of image-text pairs in a training batch; $I_i$ is the $i$-th input image and $t_i^l$ denotes its label at level-$l$; $\mathcal{V}^l(I_i)$ is the normalized embedding feature of image $I_i$ computed by the head corresponding to level $l$ (Figure 2), $\mathcal{T}(t_i^l)$ is the normalized text embedding of the label at level-$l$. While previous studies have demonstrated the effectiveness of pre-trained VLMs in standard image classification Xiao et al. (2022); Jin et al. (2021), this work explores their potential for hierarchical classification, with two main goals: 1) to quantify the performance improvements they provide, and 2) to evaluate their effectiveness in addressing the hierarchical manifold challenge.

## 4 EXPERIMENTS

We conducted thorough experiments to validate our approaches. Firstly, we confirmed that graph diffusion-based methods outperform current top-down and bottom-up inference methods in hierarchical classification (Section 4.1). Then, we demonstrated the benefits of fine-tuning with text encoders and hierarchical supervision through both qualitative and quantitative analyses (Section 4.2). Finally, we provided a clear quantitative comparison among these methods and other prominent approaches in hierarchical classification (Section 4.3).

**Datasets**. We conduct a comprehensive evaluation of hierarchical classification methods on two prominent datasets: iNaturalist18 (iNat18)Van Horn et al. (2018) and iNaturalist21-mini (iNat21)Van Horn et al. (2021). The iNat18 dataset comprises 437,500 samples from 8,142 species, while iNat21 includes 500,000 training samples from 10,000 species. It is important to note that iNat18 exhibits a long-tailed distribution, in contrast to the balanced iNat21. Both datasets are structured hierarchically with 7 levels. While the work by Valmadre (2022) focuses on hierarchical classification using iNat21, it does not include an analysis of iNat18. Our research extends this work to iNat18 to provide a more comprehensive assessment of our model's performance in the context of long-tailed data distributions.

**Metrics**. In accordance with the methodology proposed by Valmadre (2022), we employ a suite of performance metrics derived from operating curves. These include Average Precision (AP), Average Correct (AC), Recall at X% Correct (R@X). AP and AC are defined as integrals with respect to Recall. Additionally, we introduce single prediction metrics such as Majority F1 (M-F1), Leaf F1 (L-F1), and Leaf Top1 (L-Top1) Accuracy. While Leaf Top1 Accuracy provides a measure of accuracy at the leaf level, the other metrics are designed to assess the performance of hierarchical classification. Our analysis reveals that the leaf-level metric L-Top1 does not consistently align with hierarchical metrics such as AP, as demonstrated in Table 3.

## 4.1 Graph Diffusion based Inference Methods

**Comparison with other inference methods**. We evaluated our diffusion-based techniques, including both general and differentiable diffusion, against traditional top-down Redmon & Farhadi (2017); Jain et al. (2023) and bottom-up Valmadre (2022); Wu et al. (2020) inference methods. The results, presented in Table 1, reveal that our methods surpass existing ones. Intriguingly, diffusion not only enhances hierarchical metrics but also boosts the leaf-level top-1 accuracy. The leaf-level top-1 accuracy on our HMCF L1-7 (models of hierarchical multi-modal cross-modal finetuning) improves 7% by using our diffusion-based inference. Note that our general diffusion doesn't necessitate extra training, making this discovery particularly noteworthy.

**Generality on other fine-tuned models**. Our diffusion-based inference is a general and model-agnostic approach and can be used for other fine-tuned models. We show its performance on different fine-tuned models in Table 2. The models are finetuned with different training losses elaborated in Section 4.2. The result shows that our graph diffusion and its differential version consistently improve the bottom-up inference.

Table 1: Evaluation of our diffusion-based inference against SOTA methods on iNat18. We use the backbone learned by HMCF L1-7 in Table 4 for all the methods. Clearly, our diffusion and differentiable (Diff.) diffusion approaches outperform the compared methods.

| Model | AP | AC | R@90 | R@95 | M-F1 | L-F1 | L-Top1 |
|---|---|---|---|---|---|---|---|
| Top-down Redmon & Farhadi (2017) | 64.36 | 61.72 | 46.10 | 34.97 | 68.54 | 68.36 | 46.62 |
| Level-top-down Jain et al. (2023) | 72.11 | 69.98 | 58.09 | 46.96 | **76.23** | 75.96 | 55.71 |
| Bottom-up Valmadre (2022) | 72.75 | 70.60 | 59.56 | 52.60 | 72.73 | 75.16 | 55.78 |
| Diffusion | 73.48 | 71.88 | **62.48** | **55.53** | 75.94 | 75.71 | 56.33 |
| Diff. diffusion | **73.82** | **71.91** | 61.99 | 53.36 | 76.01 | **76.09** | **59.70** |

Table 2: Our diffusion (D) and differentiable diffusion (DD) inference methods improve the performance of bottom-up (BU) across all metrics and various models. We test models trained with HMCF, CE, and descendant softmax (Desc. softmax) using labels at all levels (L1-7) and at levels 6 and 7 (L67). "IN" indicate pretrained model of ImageNet. All models leverage the CLIP visual encoder as a pre-trained model, except specified with "IN".

| Model | AP | AC | R@90 | R@95 | M-F1 | L-F1 | L-Top1 |
|---|---|---|---|---|---|---|---|
| HMCF L67 BU | 72.64 | 70.65 | 60.53 | 53.22 | 72.85 | 74.88 | 56.10 |
| HMCF L67 D | **73.35** | **71.63** | **62.26** | **55.25** | 74.57 | **75.51** | 56.84 |
| HMCF L67 DD | 73.23 | 71.37 | 61.38 | 53.30 | 75.48 | 75.44 | **59.51** |
| HMCF L1-7 BU | 72.75 | 70.60 | 59.56 | 52.60 | 72.73 | 75.16 | 55.78 |
| HMCF L1-7 D | 73.60 | 71.85 | **62.06** | 54.97 | 74.79 | 75.82 | 56.50 |
| HMCF L1-7 DD | **73.82** | **71.91** | 61.99 | 53.36 | **76.01** | **76.09** | **59.70** |
| CE L67 BU | 69.18 | 67.07 | 56.32 | 48.28 | 71.99 | 71.81 | 53.68 |
| CE L67 D | **69.45** | **67.56** | **56.47** | 48.61 | **72.57** | **72.31** | **54.14** |
| CE L67 DD | 69.20 | 67.12 | 56.40 | **48.75** | 71.96 | 71.81 | 53.84 |
| Desc. softmax Valmadre (2022) BU | 58.53 | 55.86 | 40.28 | 33.71 | 58.73 | 63.50 | 45.10 |
| Desc. softmax D | **60.70** | **58.58** | **45.41** | **37.84** | **64.65** | **64.31** | **45.66** |
| Desc. softmax DD | 59.98 | 57.73 | 44.68 | 37.13 | 62.77 | 63.43 | 45.62 |
| Desc. softmax IN L1-7 BU | 65.66 | 62.81 | 46.86 | 39.73 | 62.30 | 70.12 | 51.42 |
| Desc. softmax IN L1-7 D | 66.88 | 64.84 | 52.43 | 44.28 | **69.91** | 70.02 | 51.43 |
| Desc. softmax IN L1-7 DD | **67.51** | **65.34** | **53.60** | **45.56** | 68.68 | **70.31** | **52.78** |

**Ablation study of graph diffusion parameters.** $\alpha$ and iteration $t$ are two important hyperparameters for our diffusion inference. As shown in Figure 3, the hierarchical metrics initially increase and then decrease with changes in the parameter $\alpha$. These metrics generally converge after about 4 iterations. Based on this observation, we employ $\alpha = 0.3$ and $t = 12$ in all the experiment in this paper.

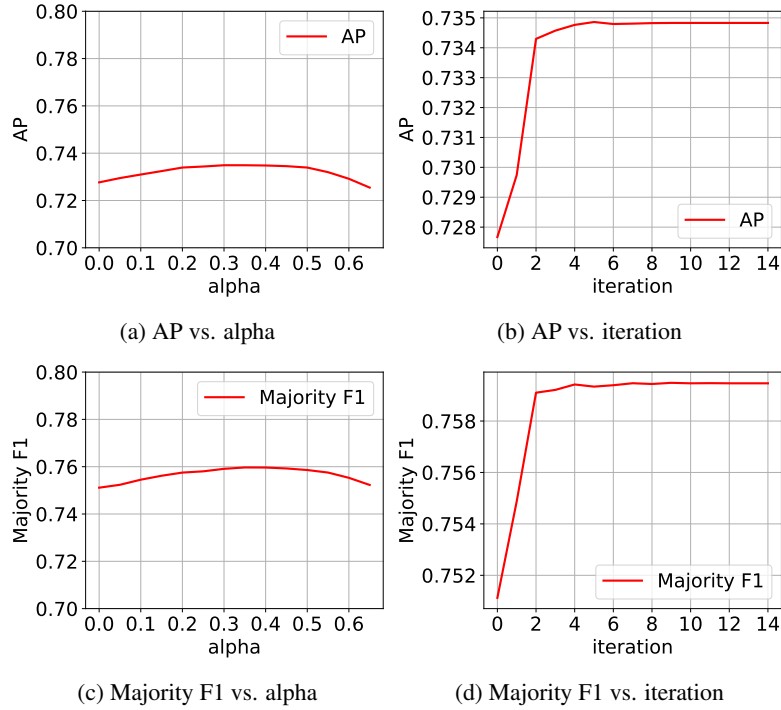

(a) AP vs. alpha  (b) AP vs. iteration

(c) Majority F1 vs. alpha  (d) Majority F1 vs. iteration

Figure 3: Ablation study of diffusion hyperparmeter $\alpha$ and the number of iterations $t$. We use the backbone learned by HMCF in Table 4 here.

## 4.2 HIERARCHICAL CLASSIFICATION FROM MANIFOLD PERSPECTIVE

To see whether hierarchical supervision and training with language models reduce manifold overlaps and improve hierarchical metrics, we have compared four distinct pipelines as shown in Table 3. Their distinction lies in the utilization of a language model, and the application of either leaf-level or level-wise hierarchical supervision during fine-tuning. It is imperative to highlight that all models are trained utilizing the same pretrained model and identical hyperparameters. The qualitative and quantitive results are shown in Figure 4 and Table 3, respectively.

Table 3: Ablation study of our HMCF on iNat18. Models trained with cross-entropy loss (CE) and multi-modal contrastive training loss (MCL) are modified to level-wise hierarchical format with different levels selected (level 7 and whole levels). The hierarchical metrics benefits from hierarchical information and multi-modal contrastive finetuning. We provide a visualization of the hierarchical manifolds of the methods in Figure 4.

| Loss | AP | AC | R@90 | R@95 | M-F1 | L-F1 | L-Top1 |
|---|---|---|---|---|---|---|---|
| Flat softmax L7 (Figure 4a) | 67.90 | 65.89 | 54.63 | 46.51 | 70.75 | 70.58 | 54.10 |
| Flat softmax L1-7 (Figure 4b) | 70.53 | 68.54 | 57.42 | 50.48 | **73.61** | 73.07 | 55.16 |
| MCF L7 (Figure 4c) | _72.40_ | _70.33_ | _59.36_ | _52.42_ | 72.33 | _74.72_ | **56.69** |
| MCF L1-7 (Ours, Figure 4d) | **72.75** | **70.60** | **59.56** | **52.60** | _72.73_ | **75.16** | _55.78_ |

**Use of text encoder and multi-modal contrastive loss**. While the effectiveness of leveraging the CLIP pre-trained encoder using contrastive loss has been previously noted in standard classification Xiao et al. (2022), we investigate the potential benefits of these models for hierarchical classification in this paper, aiming to 1) quantify the extent of performance improvement they offer and 2) verify their effectiveness in tackling the hierarchical manifold issue. We compare two kinds of training losses in this subsection: cross-entropy (CE) and multi-modal contrastive finetuning (MCF). The latter take language model for finetuning and both architectures can be modified to level-wise hierarchical version. As shown in Table 3, training with MCL outperforms CE Goyal et al. (2023). When only using the leaf level labels, the AP improves 6.6% by changing the loss from CE to MCL, indicating that fine-tuning with MCL is more effective than the traditional CE in hierarchical

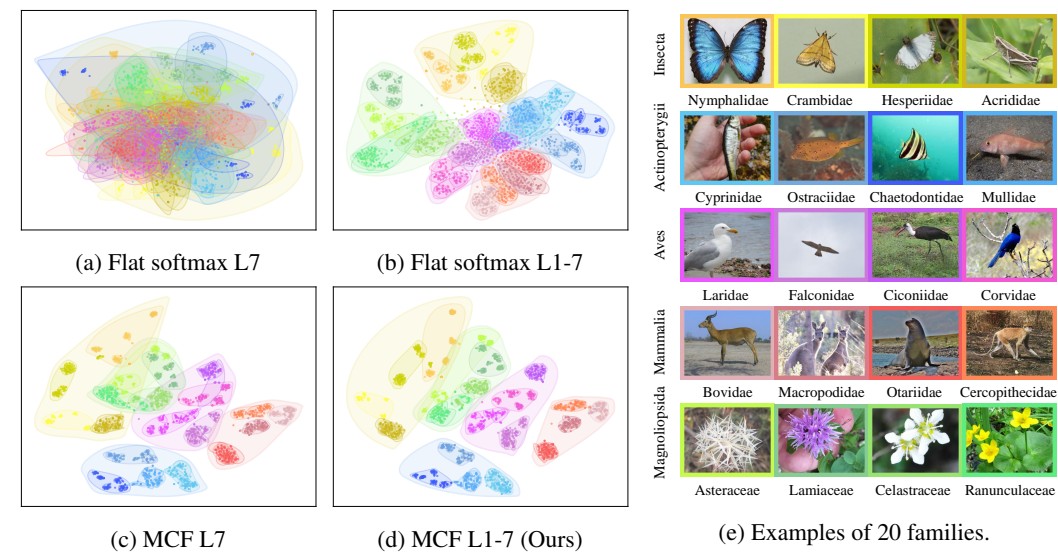

Figure 4: Visualization of 2D embedding with t-SNE of different training methods on iNat18. All models are finetuned based on CLIP ResNet50. Corresponding hierarchical metrics can be found in (Table 3). (a) Finetuning with CE loss using leaf-level labels does not separate manifolds at coarse levels although it is competitive of top-1 accuracy at the leaf level among compared methods (Table 4). (b) Finetuning with hierarchical labels and CE loss provides less overlap between coarse-level manifolds than CE loss finetuning on only leaf level. (c) Multi-modal contrastive finetuning on the leaf level provides less overlap between coarse-level manifolds than CE loss finetuning on the leaf node. (d) Hierarchical multi-modal contrastive finetuning produces less overlap between coarse-level and fine-grained level manifolds, which provides better hierarchical metrics (Table 3) than other finetuning approaches. (e) Example images in this visualization. We select five classes (level 3); for every class we select four families (level 5).

classification. We visualized the embedding features of images from different categories using t-SNE. Comparing Figure 4a and Figure 4c, it shows that MCL reduces the manifold overlap, especially at the coarse level. It shows that the CLIP text encoder and the contrastive loss are more effective than CE in dealing with hierarchical manifolds in the hierarchical classification problem. Additional qualitative and quantitative results are provided in the appendix for reference.

**Hierarchical supervision.** Table 3 shows that hierarchical supervision improves the performance of leaf-level supervision. The visualization result in Figure 4 shows that embedding features from different categories fine-tuned by hierarchical labels (Figure 4b and Figure 4d) share less hierarchical manifold overlap than only using the leaf-level supervision (Figure 4a and Figure 4c). The improvement of hierarchical supervision in CE is larger than that in MCL; using whole levels (1-7) on CE improves the AP by $3.9\%$ than using only leaf labels. Interestingly, incorporating additional levels on MCL does not consistently improve all hierarchical metrics, as shown in Table 3 (compare MCL7 and MCL1-7). Notably, these findings diverge from the prevailing belief that top-1 accuracy benchmarks align with hierarchical metric rankings Russakovsky et al. (2015), underscoring the importance of studying hierarchical metrics.

### 4.3 Comparision with SOTA Hierarchical Classification Methods

In this subsection, we showcase qualitative results comparing SOTA hierarchical classification methods Valmadre (2022) with our HMCF and diffusion. We introduce the implementation detail in the appendix.

**Compared methods**. The flat softmax classifier Bertinetto et al. (2020), even without using class hierarchies during training, is a strong baseline. The conditional softmax (Cond softmax) classifier Redmon & Farhadi (2017), known from YOLO-9000, elegantly degrades by predicting conditional distributions of child classes given their parents, while the conditional sigmoid (Cond sigmoid) Brust

& Denzler (2019) extends this to support multi-path labels in hierarchies. Multilabel focal adopts focal lossLin et al. (2017) for training. The Deep Realistic Taxonomic Classifier (Deep RTC) Wu et al. (2020) sums ancestor scores for node evaluation and is noted for its competitiveness. The Parameter Sharing (PS) softmax Wu et al. (2020), a simplification of Deep RTC that shares parameters across different parts of the hierarchy, has proved robust and effective, and the soft-max-margin loss function (Softmargin) Valmadre (2022) involves modifying the decision boundary to allow for a certain degree of misclassification. The descendant loss (Desc. softmax) Valmadre (2022) involves predicting the distribution of descendent classes in the hierarchy. These methods collectively highlight the nuanced trade-offs between specificity and generalization in hierarchical classification tasks.

Table 4: Benchmarking results on the iNat18 dataset. We report numbers w.r.t both hierarchical metrics Valmadre (2022) and the standard top-1 accuracy on leaf classes (dubbed L-Top1 in the last column). HMCF contrastively fine-tunes a pretrained model using all the taxonomic levels and outperforms prior arts. Additionally applying diffusion improves performance notably further. All the models are finetuned based on the same pre-trained CLIP ResNet50 visual encoder.

| Model | AP | AC | R@90 | R@95 | M-F1 | L-F1 | L-Top1 |
|---|---|---|---|---|---|---|---|
| Flat softmax Bertinetto et al. (2020) | 67.90 | 65.89 | 54.63 | 46.51 | 70.75 | 70.58 | 54.10 |
| Multilabel focal Lin et al. (2017) | 62.72 | 59.83 | 45.26 | 38.36 | 63.08 | 66.50 | 43.95 |
| Cond. softmax Redmon & Farhadi (2017) | 57.99 | 54.87 | 41.05 | 35.02 | 62.02 | 61.89 | 38.11 |
| Cond. sigmoid Brust & Denzler (2019) | 57.68 | 54.71 | 40.61 | 34.11 | 60.74 | 62.50 | 38.90 |
| Deep RTC Wu et al. (2020) | 68.38 | 62.84 | 31.95 | 19.56 | 73.52 | 73.59 | 56.29 |
| PS softmax Wu et al. (2020) | 70.37 | 68.60 | 58.32 | 51.25 | 72.98 | 72.82 | 56.31 |
| Desc. softmax Valmadre (2022) | 59.31 | 56.40 | 42.27 | 34.83 | 60.64 | 63.81 | 42.00 |
| Softmargin Valmadre (2022) | 66.40 | 64.07 | 53.47 | 45.51 | 69.63 | 69.48 | 53.39 |
| HMCF (Our) | 72.75 | 70.60 | 59.56 | 52.60 | 72.73 | 75.16 | 55.78 |
| HMCF + diffusion (Our) | **73.48** | **71.88** | **62.48** | **55.53** | **75.94** | **75.71** | **56.33** |

Table 5 in appendix exemplifies the hierarchical performance of mainstream methodologies and iNat21. Please note that all methods undergo fine-tuning using the same pre-trained CLIP ResNet50 visual encoder. To ensure a fair comparison, we employ identical training conditions, including the Adam optimizer and batch size until convergence is reached. The detailed analysis and exact-correct and recall-precision operating curves for each method are illustrated in appendix. Our results demonstrate that fine-tuning with the CLIP text encoder (HMCF) enhances hierarchical performance, with further improvements observed when utilizing the graph diffusion-based approach (HMCF + diffusion) in hierarchical classification. Our result on iNat18 in the appendix shows a similar trend with Table 5.

### 4.4 LIMITATIONS AND FUTURE WORK

Vision-language models and graph diffusion provide a new perspective for the long-tailed hierarchical classification tasks. Currently we apply contrastive learning with the simplistic prompt template ("a photo of a {class}") of hierarchy node names. What kind of prompt is more appropriate for hierarchical classification is a task worthy of investigating in the future. Besides, several related aspects are still worth in-depth study, such as automatic hierarchy construction, hierarchical training loss design for long-tailed benchmarks, and methods for multi-granularity aggregation. From this perspective, currently our design is primitive and we hope our work can serve as a good start point.

## 5 CONCLUSIONS

This paper introduces a new perspective on the hierarchical classification problem by viewing it through the lens of manifold learning. Leveraging this approach, we present innovative strategies for training and inference. Our proposed hierarchical multi-modal contrastive loss and graph-based diffusion methods for hierarchical predictions offer a nuanced balance between coarse and fine-class predictions. Evaluations on iNat18 and iNat21 datasets demonstrate the superior performance of our methods in terms of both top-1 accuracy and various hierarchical metrics, marking a notable advancement in the field of hierarchical classification.

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

# APPENDIX

In this appendix, we present operating curves and a comprehensive analysis of various hierarchical methods (Section A). Additionally, we include ablations of graph diffusion, which encompass variations in graph diffusion input (Section B.1), as well as implementations and evaluations of training losses for differential diffusion (Section B.2). Furthermore, we provide ablations related to training methods, including comparisons of different pretrained models (Section C.1), manifold visualizations of visual and text embeddings across various training methods (Section C.2), and assessments of different contrastive learning techniques (Section C.3). Finally, we discuss training and inference efficiency in Section D.

## A  OPERATING CURVES AND DETAILED ANALYSIS OF DIFFERENT HIERARCHICAL METHODS

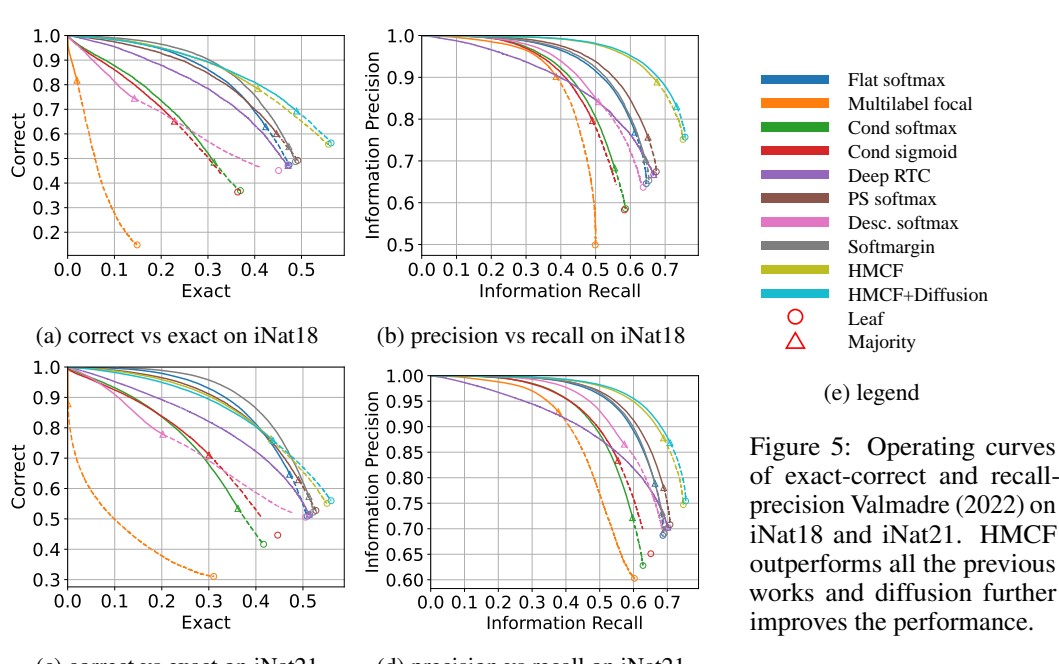

(a) correct vs exact on iNat18     (b) precision vs recall on iNat18

(c) correct vs exact on iNat21     (d) precision vs recall on iNat21

(e) legend

Figure 5: Operating curves of exact-correct and recall-precision Valmadre (2022) on iNat18 and iNat21. HMCF outperforms all the previous works and diffusion further improves the performance.

Table 5: Benchmarking results on iNat21. We report numbers w.r.t both hierarchical metrics Valmadre (2022) and the standard top-1 accuracy on leaf classes (dubbed L-Top1). Conclusions hold as in Table 4. All the models are finetuned based on the same pre-trained visual encoder.

| Model | AP | AC | R@90 | R@95 | M-F1 | L-F1 | L-Top1 |
|---|---|---|---|---|---|---|---|
| Flat softmax  Bertinetto et al. (2020) | 66.17 | 64.32 | 53.85 | 47.02 | 68.87 | 68.69 | 50.89 |
| Multilabel focal  Lin et al. (2017) | 54.58 | 50.35 | 36.16 | 30.45 | 50.62 | 60.27 | 31.05 |
| Cond softmax Redmon & Farhadi (2017) | 58.88 | 56.26 | 42.95 | 36.23 | 62.85 | 62.80 | 41.64 |
| Cond sigmoid  Brust & Denzler (2019) | 59.24 | 56.74 | 42.84 | 35.61 | 61.41 | 65.11 | 44.64 |
| Deep RTC  Wu et al. (2020) | 63.92 | 58.07 | 25.36 | 14.10 | 70.17 | 70.22 | 51.43 |
| PS softmax  Wu et al. (2020) | 68.22 | 66.49 | 56.20 | 49.85 | 71.07 | 70.80 | 52.76 |
| Desc. softmax  Valmadre (2022) | 64.95 | 62.71 | 48.84 | 42.59 | 64.64 | 69.03 | 50.55 |
| Softmargin  Valmadre (2022) | 66.53 | 64.72 | 54.41 | 47.91 | 69.39 | 69.09 | 52.22 |
| HMCF (Our) | 72.46 | 70.52 | 60.49 | 53.66 | 73.35 | 74.72 | 55.11 |
| HMCF + diffusion (Our) | **73.16** | **71.62** | **62.81** | **55.97** | **75.31** | **75.32** | **55.86** |

**Implementation detail for fair comparison**. Figure 5 and Table 5 show the operating curves and quantitive comparision on iNat21. We follow the explored training configurations by Valmadre Valmadre (2022) in implementing the SOTA methods. During fine-tuning, the learning rate follows a

cosine function with maximum value $1 \times 10^{-5}$. We use the AdamW optimizer with weight decay $1 \times 10^{-1}$. Models are trained on a single A100 with batch size 64 for 100 epochs. For each model, we train it three times and report their average top-1 accuracy. The standard deviation for all the methods is less than 0.3% in accuracy, which is sufficiently small to draw conclusions. Notably, all models are trained utilizing the identical pretrained CLIP ResNet50 He et al. (2016) visual encoder.

**Detailed analysis of training**. Parameter sharing plays a crucial role in hierarchical classification, which is also adopted in our proposed HMCF. As shown in Table 4, Deep RTC Wu et al. (2020) and its variance PS softmax get relatively high metrics such as AP or M-F1. Deep RTC utilizes parameter sharing through a shared backbone feature extractor across all label sets. Its predictor reflects a hierarchical architecture, enabling it to achieve high hierarchical metrics such as M-F1 (73.52). However, it achieves a relatively low recall (e.g., R@95 is only 14.10) due to its preference for coarse predictions. PS softmax Wu et al. (2020) improves upon Deep RTC Wu et al. (2020) by learning a linear reparametrization from parameter sharing scores, resulting in improved hierarchical metrics overall. Parameter sharing connects the knowledge of coarse and fine-grained semantic levels in hierarchy. We found training with text encoder fulfills the requirements for parameter sharing. First, it meticulously design a multi-branch architecture, facilitating knowledge sharing between coarse and fine-grained levels while generating embeddings of different levels. Second, the visual encoder is supervised with a shared weighted text encoder for all nodes in the hierarchy, thereby optimizing the utilization of hierarchical information in the text encoder. Please note that the text encoder naturally contains hierarchy information, but it is not enough for hierarchical classification on its own. This information is further improved during fine-tuning, and the comparison of text embeddings visualization can be found in the appendix.

**Analysis of training loss.** Most models in Table 4 are trained using cross-entropy loss (CE), our research shows that multi-modal contrastive loss produces better hierarchical metrics (Table 3). For example, Flat softmax Bertinetto et al. (2020) uses cross-entropy for training and achieves impressive hierarchical metrics (e.g., 67.9% AP). However, it only focuses on the distinguishability of leaf-level manifolds, overlooking middle or coarse levels. Further experiments shows hierarchical supervision during training improves hierarchical metrics by optimizing manifolds across different levels of the hierarchy. Further details are available in Section 4.2.

**Analysis of inference.**. Graph diffusion-based inference methods effectively integrates prediction results across different hierarchy levels, leading to high hierarchical performance (Table 4 or Table 2). During inference, mid-level predictions can also be used for leaf-level score adjustment. Flat softmax Bertinetto et al. (2020) applies inference directly with leaf-level predictions without score adjustment. Deep RTC Wu et al. (2020) performs inference by greedy top-down traversal, which may cause error accumulation. Treating the hierarchy as a connection matrix, graph diffusion-based inference creates direct connections between hierarchy nodes, leading to improved hierarchical performance. Details in Section 4.1.

# B ABLATIONS OF DIFFUSION

## B.1 ABLATIONS OF INPUT OF DIFFUSION

Table 6: Ablation study focusing on the influence of diffusion inputs. we observed that restricting diffusion application to only level 7 (L7) yields marginal improvements. Conversely, extending diffusion to encompass additional levels, specifically levels 6 and 7 (L67) as well as levels 1 through 7 (L1-7), results in clear enhanced performance.

| Model | AP | AC | R@90 | R@95 | M-F1 | L-F1 | L-Top1 |
|---|---|---|---|---|---|---|---|
| HMCF | 72.75 | 70.60 | 59.56 | 52.60 | 72.73 | 75.16 | 55.78 |
| +Diffusion L7 | 72.79 | 70.95 | 60.56 | 53.75 | 75.12 | 75.18 | 55.76 |
| +Diffusion L67 | 73.27 | 71.59 | 61.88 | 54.80 | 75.71 | 75.58 | 56.24 |
| +Diffusion L1-7 | **73.48** | **71.88** | **62.48** | **55.53** | **75.94** | **75.71** | **56.33** |

In this subsection, we present an ablation study of diffusion input. Our findings demonstrate that increased diffusion with more hierarchy levels, incorporating more coarse-level information, leads to

improved hierarchical metrics. Additionally, We analyze the effect of truncating low scores in the diffusion input.

**Ablation study of input of graph diffusion.** The input of diffusion, denoted as $f_0 \in R^n$, where $n$ represents the number of categories in the taxonomy, corresponds to the initial output of the fine-tuned network. We can either use only the initial scores for leaf-level nodes and set the others as zero, or utilize the initial predictions for all nodes in the taxonomy tree. In our investigation presented in Table 6, we explore the impact of different types of diffusion inputs. The performance progressively improves with the inclusion of more hierarchy levels, suggesting that hierarchical performance benefits from additional mid-level information.

**Truncation of diffusion input scores.** Truncation, a well-known technique in diffusion, involves using only the top $N$ category scores as the input of diffusion to mitigate the negative influence of low-probability categories. In our ablation study, we explore the impact of truncation in diffusion. We use scores of HMCF with levels 6 and 7 as diffusion input, and vary the truncation parameter N from 8142 to 1 for multiscale analysis. As shown in Table 7, the results indicate that as $N$ decreases, M-F1 increases while other hierarchical metrics such as AP, AC, R@90, and R@95 decrease. This suggests that M-F1 benefits slightly from truncation, while other metrics do not exhibit similar improvement.

Table 7: Truncation of low scores before diffusion. The diffusion input comprises the top $N$ nodes of each hierarchy level, while all other low scores are adjusted to zero. As the value of N decreases, the hierarchical metrics (AP, AC, R@90, R@95) decline, highlighting the significance of low scores in the diffusion input for these metrics. The M-F1 score reaches its maximum when the reserved number $N$ is set to 3, suggesting that M-F1 benefits from truncation. HMCF L67 is used in this experiment.

| Reserved | AP | AC | R@90 | R@95 | M-F1 | L-F1 | L-Top1 |
| --- | --- | --- | --- | --- | --- | --- | --- |
| 8142 | **72.69** | **70.84** | **61.20** | 53.84 | 74.08 | 74.90 | 56.12 |
| 1000 | 72.69 | 70.84 | 61.20 | **53.85** | 74.08 | 74.90 | 56.12 |
| 100 | 72.67 | 70.83 | 61.16 | 53.81 | 74.16 | 74.90 | 56.13 |
| 10 | 72.51 | 70.68 | 61.06 | 53.34 | 74.62 | 74.91 | 56.16 |
| 5 | 72.20 | 70.24 | 60.44 | 52.68 | 74.92 | 74.90 | 56.17 |
| 3 | 71.49 | 69.13 | 59.96 | 49.49 | **75.03** | **74.91** | **56.18** |
| 2 | 70.11 | 67.03 | 57.02 | 0.00 | 74.88 | 74.88 | 56.18 |
| 1 | 62.38 | 55.94 | 0.00 | 0.00 | 74.87 | 74.87 | 56.10 |

### B.2 Implementations and Ablations of Differential Diffusion

In the context of differentiable diffusion, we train a bias-free linear classifier that transforms features from each hierarchy level to leaf scores for metric calculation. Specifically, we exclusively utilize the output scores of leaf classes for hierarchical metric computation (where mid-level node scores are obtained by summing the scores of their leaf descendants). Additionally, we propose training the linear classifier using both cross-entropy loss and restraint loss.

As indicated in Table 8, we use restraint loss to restrain wrongly predicted scores with high confidence. The hierarchical cross-entropy loss is defined as:

$$\mathcal{L}_{CE} = \sum_l \left( -\alpha^l \sum_k \left( y_k^l \log s_k^l \right) \right) \tag{7}$$

where $y_k^l$ and $s_k^l$ are ground truth and predicted scores of category $k$ at level $l$, separately. Hierarchical cross-entropy loss is the weighted sum of the cross-entropy loss at all hierarchy levels with weights $\alpha^l$. Scores of mid-level nodes are calculated by the sum of their leaf descendants. Hierarchical restraint loss is defined as follows:

$$\mathcal{L}_R = \sum_l \left( -\beta^l \max_k \left( (1 - y_k^l) \log(1 - s_k^l) \right) \right) \tag{8}$$

We identify the wrong-predicted nodes with the highest probability at each level and calculate the hierarchical restraint loss as a weighted sum of their losses, level by level, using weights $\beta^l$. For iNat18, we assign values of 10, 5, 3, 1, 0.5, 0.2 from level 1 to level 6 (where level 7 represents the

leaf level). The training involves the sum of hierarchical cross-entropy loss and hierarchical restraint loss.

We analyze the impact of the proposed restraint loss in Table 8. Logits are obtained from ResNet50 trained with Hierarchical Multi-modal Contrastive Loss (HMCF), and we compare two baselines: HMCF with levels 6 and 7, and HMCF with all levels. Results in Table 8 demonstrate that training a mapping matrix with cross-entropy (CE) loss can improve hierarchical metrics significantly compared to baselines, especially for L-Top1. For instance, comparing HCCF L67 CE with HCCF L67 or HCCF L1-7 CE with HCCF L1-7. Additionally, incorporating both cross-entropy and restraint loss further enhances performance, as shown in comparisons like HCCF L67 CE+R with HCCF L67 CE or HCCFL1-7 CE+R with HCCF L1-7 CE. Training with logits of all levels produces superior results compared to using only levels 6 and 7. It is worth noting that when evaluating baseline performance, only leaf (level 7) scores are considered for metric calculation. Differential diffusion acts as a consolidation of all predicted nodes in the hierarchy, proving to be a straightforward and effective method for improving both leaf-level and hierarchical metrics.

Table 8: Performance of differential diffusion with or without restraint loss on iNat18. Two baseline models are presented here: ResNet50 trained with hierarchical cross-modal contrastive learning at levels 6 and 7 (L67), and with all levels (L1-7). "CE" denotes cross-entropy loss, and "R" denotes restraint loss. This table highlights three key points: (1) The differential diffusion improves hierarchical performance. (2) Feeding more levels of logits produces better hierarchical metrics when using differential diffusion. (3) Training differential diffusion matrix with restraint loss further amplifies performance. Details in Section B.2

| Base | CE | R | AP | AC | R@90 | R@95 | M-F1 | L-F1 | L-Top1 |
|------|-----|-----|-------|-------|-------|-------|-------|-------|--------|
| L67 | - | - | 72.64 | 70.65 | 60.53 | 53.22 | 72.85 | 74.88 | 56.10 |
| L67 | ✓ | - | 72.94 | 71.77 | 60.91 | **53.74** | 75.27 | 75.18 | 59.34 |
| L67 | ✓ | ✓ | 73.23 | 71.37 | 61.38 | 53.30 | 75.48 | 75.44 | 59.51 |
| L1-7 | - | - | 72.75 | 70.60 | 59.56 | 52.60 | 72.73 | 75.16 | 55.78 |
| L1-7 | ✓ | - | 73.58 | 71.65 | 61.62 | 52.86 | 75.73 | 75.87 | 59.59 |
| L1-7 | ✓ | ✓ | **73.82** | **71.91** | **61.99** | 53.36 | **76.01** | **76.09** | **59.70** |

# C ABLATIONS OF TRAINING METHODS

## C.1 COMPARISION OF PRETRAINED MODELS

Table 9 compares the hierarchical metrics of the flatsoftmax of ResNet50 with different pretrained models, ImageNet and CLIP. The CLIP pretrained model exhibits slightly better performance than the ImageNet pretrained model. Additionally, both models benefit from hierarchical supervision. Besides,

Table 9: Hierarchical metrics of models trained based on ImageNet (IN) and CLIP pretrained model on iNat18. Models are trained with cross-entropy loss on Leaf level (L7) and whole levels (L1-7). CLIP pretrained model performs slightly better than ImageNet for hierarchical metrics and both of them benefit from hierarchical supervision.

| Pretrained Model | AP | AC | R@90 | R@95 | M-F1 | L-F1 | L-Top1 |
|------------------|-------|-------|-------|-------|-------|-------|--------|
| IN L7 | 66.24 | 64.21 | 52.40 | 44.21 | 69.38 | 69.22 | 52.31 |
| CLIP L7 | 67.90 | 65.89 | 54.63 | 46.51 | 70.75 | 70.58 | 54.10 |
| IN L1-7 | 67.04 | 64.79 | 51.58 | 43.41 | 71.02 | 70.31 | 51.51 |
| CLIP L1-7 | 70.53 | 68.54 | 57.42 | 50.48 | 73.61 | 73.07 | 55.16 |

## C.2 VIUALIZATION OF TEXT EMBEDDINGS

We employ t-SNE to visualize the text embedding, as shown in Figure 6, offering insight into the function of text embedding for hierarchical classification.

**Zero Shot CLIP**. We visualize the embedding features of the pre-trained CLIP model without fine-tuning. The text embedding generated by the text encoder of CLIP is employed as the weights of

Table 10: Hierarchical metrics of CLIP pretrained model (Zero Shot), finetuning linear classifier while fixing CLIP visual encoder (CE fix backbone), finetuning both visual encoder and linear classifer with CE (CE), multi-modal contrastive learning using leaf level (MCF), and hierarchical multi-modal contrastive learning with all hierarchical levels (HMCF). Pretrained models of all experiments are CLIP ResNet50 He et al. (2016). Details at Section C.2

| Model | AP | AC | R@90 | R@95 | M-F1 | L-F1 | L-Top1 |
|---|---|---|---|---|---|---|---|
| Zero Shot | 27.88 | 25.79 | 16.49 | 11.73 | 35.37 | 30.54 | 3.41 |
| CE fix backbone | 61.53 | 59.21 | 47.64 | 41.15 | 63.47 | 64.40 | 44.10 |
| CE | 67.90 | 65.89 | 54.63 | 46.51 | 70.75 | 70.58 | 54.10 |
| MCF | 72.40 | 70.33 | 59.36 | 52.42 | 72.33 | 74.72 | **56.69** |
| HMCF | **72.75** | **70.60** | **59.56** | **52.60** | **72.73** | **75.16** | 55.78 |

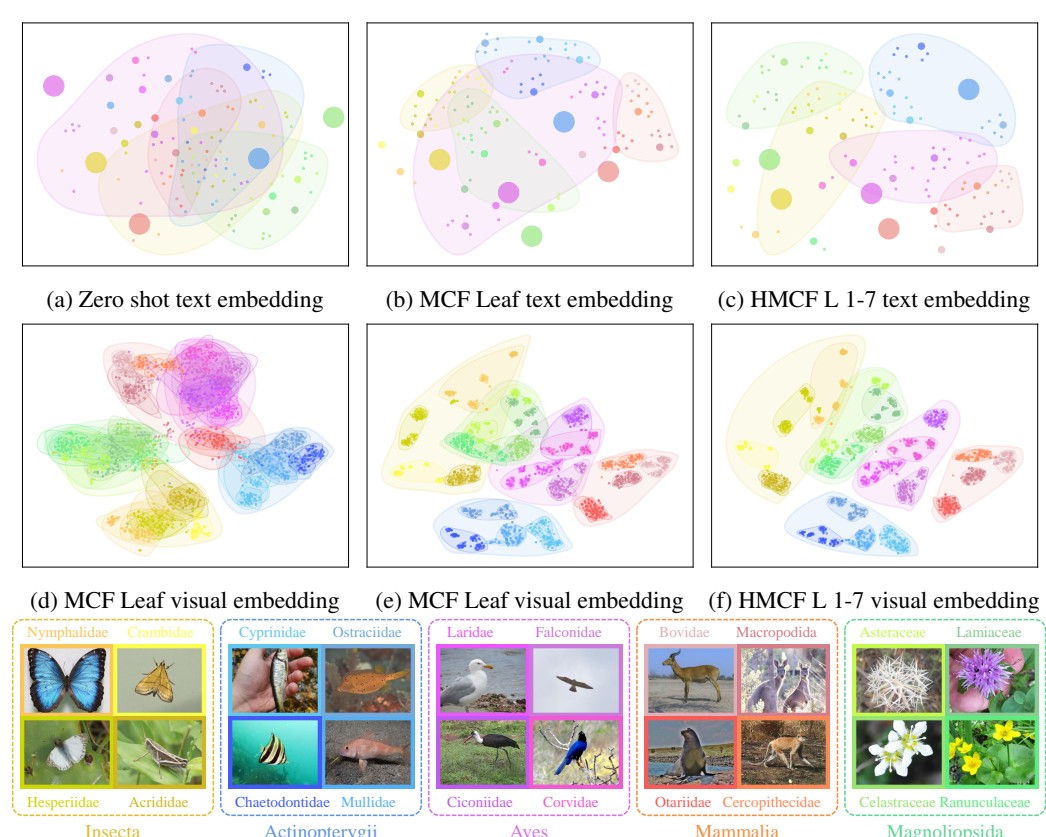

(a) Zero shot text embedding    (b) MCF Leaf text embedding    (c) HMCF L 1-7 text embedding

(d) MCF Leaf visual embedding    (e) MCF Leaf visual embedding    (f) HMCF L 1-7 visual embedding

Insecta — Nymphalidae, Crambidae, Hesperiidae, Acrididae
Actinopterygii — Cyprinidae, Ostraciidae, Chaetodontidae, Mullidae
Aves — Laridae, Falconidae, Ciconiidae, Corvidae
Mammalia — Bovidae, Macropodida, Otariidae, Cercopithecidae
Magnoliopsida — Asteraceae, Lamiaceae, Celastraceae, Ranunculaceae

(g) Family examples. 5 biological classes (level 3) consisting 20 families (level 5) are selected.

Figure 6: Visualization of t-SNE for text embeddings (subplot a, b, and c) and their corresponding visual embeddings (subplot d, e, and f). For the text embeddings (subplot a, b, and c), three different point sizes represent three hierarchy levels: large for level 3 (class), medium for level 5 (family), and small for level 7 (species). Zero-shot CLIP struggles to achieve good performance primarily due to the disorder of text embeddings (a), while the pretrained CLIP visual encoder can capture manifold at coarse levels but struggles at fine-grained levels (d). Training with multi-modal contrastive loss results in a more distinct differentiation of manifolds for both fine-grained and coarse levels (b and e). Fine-tuning with hierarchical supervision diminishes the overlap area of coarse-level manifolds for both text and visual embeddings (as shown in subplot c and f). Quantitative results are available in Table 10 and a detailed analysis is provided in Section C.2.

a linear classifier, which generates logits for each class. As shown in Figure 6a, 6d and Table 10, The pre-trained CLIP visual encoder Radford et al. (2021) demonstrates the ability to distinguish certain

coarse-level categories such as Aves (Purple points) and Actinopterygii (Blue points), although it exhibits low performance on the fine-grained L-Top1 (Table 10).

**Linear Finetuning**. We evaluate the performance of the CLIP visual encoder by keeping it fixed and training a bias-free linear classifier, initialized with the text embedding of each leaf category from iNat18 Van Horn et al. (2018). This approach leads to significant improvements in both leaf and hierarchical metrics. Further enhancing the performance, fine-tuning both the backbone and linear classifier yields additional advancements (Table 10).

**Multi-modal Contrastive Finetuning (MCF)**. The t-SNE results of MCF text and visual embeddings are depicted in Figure 6b and Figure 6e, respectively. Intriguingly, following the fine-tuning of CLIP visual and text encoders together with CLIP loss Goyal et al. (2023), both visual and text embeddings demonstrate improved manifold classification capabilities. Specifically, the overlap area between manifolds across categories at fine-grained and coarse-grained levels is notably reduced.

**Hierarchical Multi-modal Contrastive Finetuning (HMCF)**. The t-SNE results of HMCF text and visual embeddings are presented in Figure 6c and Figure 6f, respectively. HMCF notably reduces the overlap area among coarse-level manifolds compared to MCF, particularly within the text embeddings manifolds. This improvement contributes to achieving the best hierarchical metrics, as shown in Table 10.

In summary, HMCF guarantees alignment between image and text embedding with latent semantic distances. Additionally, the finetuning of the text encoder at both fine-grained and coarse-grained levels updates the distances among classes and semantic levels. Both quantitative (refer to Table 10) and qualitative (see Figure 6) results affirm that supervision with self-adapting semantic distance promotes hierarchical classification from the perspective of manifold classification.

## C.3 ABLATIONS OF CONTRASTIVE LEARNING

Table 11: Hierarchical performance of models trained with cross-entropy loss (CE), contrastive loss (SupCon), and multi-modal contrastive training loss (MCL) with or without hierarchical supervision (denoted as L7 and L1-7) on iNat18. The hierarchical metrics show the advantages of incorporating contrastive learning, multi-modal fine-tuning, and hierarchical supervision. The corresponding visualization of manifolds is presented in Figure 7. For a comprehensive explanation, refer to Section C.3.

| Loss | AP | AC | R@90 | R@95 | M-F1 | L-F1 | L-Top1 |
|------|------|------|------|------|------|------|------|
| CE L7 Bertinetto et al. (2020) | 67.90 | 65.89 | 54.63 | 46.51 | 70.75 | 70.58 | 54.10 |
| CE L1-7 Valmadre (2022) | 70.53 | 68.54 | 57.42 | 50.48 | 73.61 | 73.07 | 55.16 |
| SupCon L7 Khosla et al. (2020) | 68.70 | 66.74 | 55.98 | 48.79 | 71.46 | 71.22 | 54.03 |
| SupCon L1-7 Zhang et al. (2022) | 70.32 | 67.96 | 57.65 | 50.24 | 72.69 | 72.42 | 54.41 |
| MCL L7 Goyal et al. (2023) | 72.40 | 70.33 | 59.36 | 52.42 | 72.33 | 74.72 | **56.69** |
| MCL L1-7 (Our) | **72.75** | **70.60** | **59.56** | **52.60** | **72.73** | **75.16** | 55.78 |

We conducted an ablation study to analyze the impact of components in hierarchical multi-modal contrastive fine-tuning (HMCF): contrastive learning, multi-modal supervision, and hierarchical supervision. Results show gradual improvements compared to cross-entropy (CE) methods, supervised contrastive learning (SupCon), and multi-modal contrastive fine-tuning (MCF). Adding hierarchical information for supervision during finetuning further enhances hierarchical performance. All models were fine-tuned on iNat18 using CLIP ResNet50 as the pretrained model with 7 biological levels. We visualized manifolds using t-SNE at three hierarchy levels: class (level 3), family (level 5), and species (level 7). Well-trained models exhibit reduced interaction areas among classes at both fine-grained and coarse levels. Refer to Figure 7 for visualization and Table 11 for hierarchical performance metrics. This analysis provides a comprehensive understanding of the effectiveness of different training components in HMCF.

**Contrastive learning helps hierarchical metrics.** The cross-entropy (CE) loss separates classes equally, while supervised contrastive learning aims to reduce distances within class samples and increase gaps between classes Khosla et al. (2020). We further investigated their impact on hierarchical classification. Comparing Figure 7a and Figure 7b, training with contrastive loss results in less

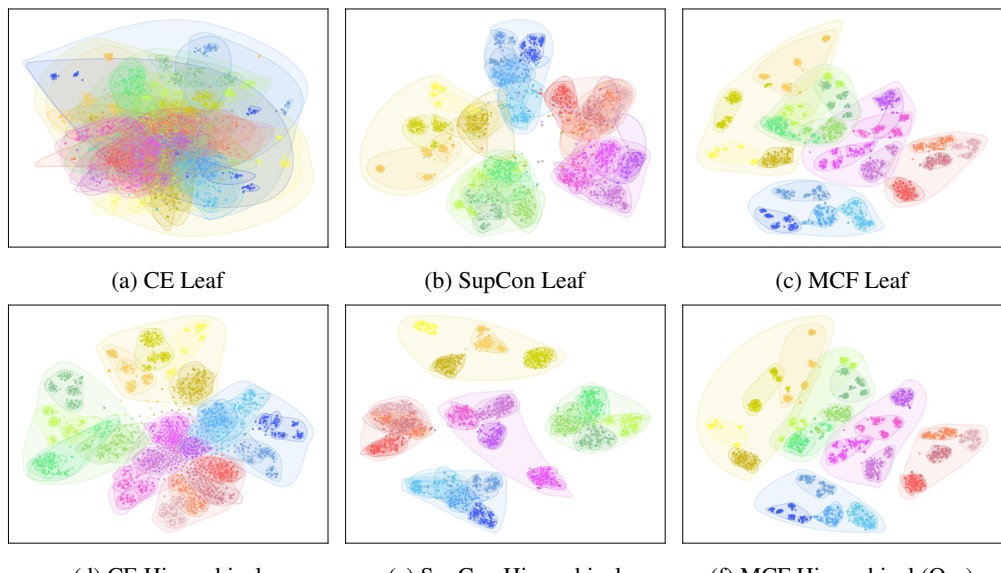

(a) CE Leaf  (b) SupCon Leaf  (c) MCF Leaf

(d) CE Hierarchical  (e) SupCon Hierarchical  (f) MCF Hierarchical (Our)

Figure 7: Visual representations of manifolds depicting 3 hierarchy levels (class, family, and species) from different training methods on iNat18 using t-SNE. This demonstrates that our proposed HMCF benefits from contrastive learning, multi-modal fine-tuning, and hierarchical supervision (Comprehensive analysis is provided in Section C.3). Subplots (a) and (d) are generated by models fine-tuned with cross-entropy loss (CE), (b) and (e) from supervised contrastive loss (SupCon), and (c) and (f) from cross-modal contrastive loss (MCF). Subplots (a), (b), and (c) only utilize leaf level information, while (d), (e), and (f) are fine-tuned with hierarchical supervision, utilizing information across all 7 levels in the hierarchy. All the experients take CLIP ResNet50 visual encoder as pretrained model. Corresponding quantitative results are in Table 11, and example visualization can be found in Figure 6g.

chaotic manifolds than CE for both fine-grained and coarse levels, leading to improved hierarchical performance (Table 11). For example, the average precision (AP) increases from 67.9 to 68.7.

**Multi-modal learning with hierarchical semantic promotes hierarchical performance.** Fine-tuning the CLIP visual and text encoders together with contrastive loss has been shown to be beneficial for downstream tasks Goyal et al. (2023). When comparing Figure 7b and Figure 7c, we observe that adding semantic information for hierarchical training reduces overlaps among fine-grained manifolds, leading to improved hierarchical performance, such as boosting the average precision (AP) from 67.52 to 72.40 (Table 11). Several factors contribute to this performance improvement. Firstly, the natural world provides abundant semantic information implicitly containing hierarchy-related semantics, as seen in the manifolds of the CLIP text encoder (refer to Figure 6). Secondly, cross-modal contrastive learning aligns visual and text embeddings, providing implicit and adaptive distance constraints for the visual encoder's learning process. Lastly, updating the text encoder during training adds flexibility to the supervision of the visual encoder's embeddings. While the effectiveness of leveraging the CLIP pre-trained encoder has been noted in contexts like few-shot classification Xiao et al. (2022) and object detection Jin et al. (2021), our work stands out as the first to apply this technique to hierarchical classification.

**Hierarchical supervision helps hierarchical metrics.** Let's compare Figure 7 vertically. Training with hierarchical supervision results in improved hierarchical manifolds for: Cross-entropy (CE) (Figure 7a vs. Figure 4d), Contrastive learning (Figure 7b vs. Figure 7e), and Cross-modal contrastive learning methods (Figure 4c vs. Figure 7f). Hierarchical supervision enhances the average precision (AP) by 3.87%, 2.36%, and 0.48% respectively (Table 11). CE with hierarchical supervision Valmadre (2022) involves increasing the output dimension of the final fully connected layer during fine-tuning. SupCon with hierarchical supervision Zhang et al. (2022) considers hierarchical distances between different fine-grained classes during contrastive learning. Our hierarchical multi-modal contrastive fine-tuning refines the visual encoder to generate level-wise visual embeddings, while sharing the same

text encoder across all hierarchical taxonomies for knowledge sharing and computational efficiency. Observing Figure 7, hierarchical supervision notably reduces overlap among coarse-level manifolds compared to leaf-level supervision. Additionally, according to Table 11, our proposed HMCF outperforms other methods in hierarchical metrics, ranking first for both leaf-top and hierarchical metrics after utilizing graph diffusion-based inference.

**Hierarchical metrics is not always consistent with leaf accuracy.** Interestingly, adding more levels in MCL does not consistently enhance all hierarchical metrics, as indicated in Table 11 (compare MCFL7 and MCFL1-7). These results challenge the common belief that top-1 accuracy benchmarks correlate with hierarchical metric rankings Russakovsky et al. (2015), emphasizing the significance of studying hierarchical metrics.

**Latent function of level-wise supervision.** Regarding models trained with CE (Figure 7d) and CMF (Figure 7f) with hierarchical supervision, they can independently generate level-wise likelihoods instead of solely leaf-level predictions. These scores are advantageous for downstream optimization, such as ensemble learning or graph diffusion-based inference. The above experiments confirm that integrating fine-grained and coarse-level information leads to improved hierarchical performance.

## D EFFICIENCY

We report training and inference time of hierarchical cross-modal contrastive learning and diffusion here.

**Training**. Training with contrastive loss requires 107.2 hours, which is longer than cross-entropy loss (52.55 hours) for 100 epochs with a batch size of 64 on a single A100 GPU.

**Inference**. The iNat18 dataset consists of 14,036 nodes (including the root node) representing 8,142 classes, requiring a 14,036x14,036 matrix for graph diffusion. Moreover, the visual encoder (ResNet50) processes images in 3.19ms per image for inference, with a slight increase to 3.27ms when diffusion is incorporated. This uptick represents just a 2.5% rise in the inference time, highlighting the efficiency of our proposed diffusion-based inference method.

