# OpenReview forum: "Hierarchical Classification via Diffusion on Manifolds"
_ICLR.cc/2025/Conference — ICLR 2025 Conference Withdrawn Submission_

### Official Review · Reviewer_TKSU · 2024-10-29

**Soundness:** 2
**Presentation:** 2
**Contribution:** 2
**Rating:** 5
**Confidence:** 3

**Summary:**

The paper proposes a combination of a contrastive fine-tuning technique and a graph diffusion inference strategy to improve the downstream performance of hierarchical classification models. This is investigated across two main benchmarks and several settings.

**Strengths:**

- The paper reads reasonably well, and the narrative is coherent.
- The experimental setup in comprehensive.
- The results look promising compared to the considered baselines.

**Weaknesses:**

- [Narrative] Some key points are repeatedly mentioned over and over again throughout the paper (eg. top-down, bottom-up, why they work and don't and why one outperforms the other and even why the proposed approach is better). Overall, the paper can be simplified/shortened to go straight to the point. I understand these are core points to take home, but stating them so many times might not be helpful nor efficient. While on the same topic, several grammatical errors and typos can be found throughout the paper, please make sure you give it another good proof-read.
- [Relevance] Most baselines and references upon which the problem definition and its significance are built date back to 2 to 5 years ago. Whether hierarchical classification is still such an important problem can be questioned in that light. My suggestion would be to definitely look for more recent literature and baselines that accentuate on the relevance and importance of the problem.
- [Proof] I'd suggest adding a proof that walks the reader from eq. (2) to (4). It is not that straightforward.
- [General] I feel using "graph diffusion" might be confusing as most recently there is a new branch in literature that focuses on using Diffusion models (in computer vision) over graphs. I'm not sure though how this could be addressed, please give it a thought.

**Questions:**

- [Methodology] It is not clear how sections 3.2 (inference), and 3.3 (fine-tuning) come together in practice. Are these both applied? Do you first finetune using HMCF, and then apply the graph diffusion? Please elaborate this, perhaps in a pseudo code, algorithmic view.
- How do you justify this swing of outperformance between D and DD across performance eval tables? Wouldn't it be possible to have one approach (a hybrid) that works best in most/all scenarios? If not, when to use which method? Please elaborate on this.

---

### Official Review · Reviewer_EE4k · 2024-11-01

**Soundness:** 3
**Presentation:** 2
**Contribution:** 2
**Rating:** 5
**Confidence:** 3

**Summary:**

This paper proposes a hierarchical multi-modal contrastive for fine-tuning pre-trained vision language models. The authors claim the method can capture hierarchy more efficiently than top-down and bottom-up approaches. Furthermore, they introduce a graph diffusion algorithm for inference motivated by a hierarchical manifold. In the end, they compare the performance of their inference and fine-tuning strategy across different tasks and datasets with other methods and show the superiority of their method compared to them.

**Strengths:**

* The paper is well written.
* The idea of hierarchical classification is exciting and important, especially in the application.
* The inference strategy based on graph diffusion is novel for this task.

**Weaknesses:**

* The discussion from lines 171-176 is confusing and not accurate since it is about the importance of incorporating the hierarchical nature of data in training and inference. Yet one of the paper's main contributions is an inference method to improve performance.
* The paper lacks cohesion. It proposes two methods that are hardly connected. One is an inference method based on graph diffusion, and the other is a fine-tuning method for VLMs based on contrastive loss.
* The numbers in all Tables don't have confidence intervals, so it is hard to grasp how significant the differences are. The authors should include confidence intervals or standard deviations from multiple runs.

**Questions:**

* What is the difference between differentiable diffusion (DD) and diffusion (D)? Since the explanation for DD is in section 3.2, the diffusion-only case needs to be explained.
* Based on the results in Tables 1 and 2, DD hardly affects AP and metrics for hierarchical classification, although it improves leaf classification. Why is this the case?
* Based on the results in Table 3, most of the improvements are due to adding MCF on leaf classes and a minor improvement by adding hierarchy. Can you explain why this is the case? Since there is a discussion in the paper that fine-tuning just on the leaf level is not effective.

Suggestions:

* Adding more discussion to connect the two methods would be great.
* It would be nice to show detailed performance based on each class to see the method's effectiveness in cases where distinguishing the hierarchical classes is more challenging.

---

### Official Review · Reviewer_TAbu · 2024-11-03

**Soundness:** 2
**Presentation:** 2
**Contribution:** 2
**Rating:** 3
**Confidence:** 3

**Summary:**

The paper addresses the challenge of classifying images within a hierarchical labels, which prioritizes correct coarse labels when fine labels are difficult to predict accurately. The authors argue that standard fine-tuning approaches, which typically optimize models on fine classes using cross-entropy loss, may not fully leverage hierarchical structures. They propose the use of a graph diffusion-based inference strategy. The approach adjusts posterior probabilities across hierarchical levels, which differs from traditional top-down and bottom-up methods by treating the hierarchical taxonomy as a graph.

**Strengths:**

This paper introduces a novel perspective on hierarchical classification by leveraging manifold learning through a graph diffusion-based inference approach. The combination of hierarchical multi-modal contrastive fine-tuning (HMCF) and graph diffusion is a creative.

**Weaknesses:**

1. The paper could improve by including a more comparison with alternative hierarchical inference strategies beyond traditional top-down and bottom-up methods.
2. the method depends on vision language models to fully leverage the hierarchical structure.
3. There is not enough details on the graph diffusion process, and it should give more intuitive explanation of the mechanism.

**Questions:**

The paper uses a generic prompt structure (“a photo of a {class}”) for training the VLM with HMCF. Did you test alternative prompt formulations?

---

### Official Review · Reviewer_Axgi · 2024-11-04

**Soundness:** 2
**Presentation:** 2
**Contribution:** 2
**Rating:** 3
**Confidence:** 4

**Summary:**

This paper proposes a training and inference method to address hierarchical classification, based on the assumption of hierarchical manifolds. First, a graph diffusion-based approach is introduced, utilizing predictions at all levels, which differs from traditional bottom-up/top-down inference methods. Next, hierarchical multi-modal contrastive finetuning is proposed. The approach demonstrated good performance on the iNat18/21 dataset, and various ablation studies were conducted to validate the proposed method.

**Strengths:**

- Hierarchical classification is a significant problem.
- The inference method using graph diffusion is interesting.
- Various ablation studies were provided to demonstrate the effectiveness of the proposed method.

**Weaknesses:**

**Clarity**

1. Figure 1 is difficult to understand. Please provide more explanation. Also, is it a visualization of actual features? If so, how was it created? If it’s intended as an intuitive example, not actual data, what evidence supports this specific representation?

2. While the hypothesis of hierarchical manifolds serves as a key motivation for the proposed method (Lines 165-169 & Figure 1), it is difficult to find strong evidence supporting this hypothesis. Specifically, why do pretrained features lie on a hierarchical manifold? I believe the structure of the feature space is significantly influenced by the training process, including factors such as the choice of loss function, model architecture, and data labeling. These factors can lead to feature spaces that do not strictly conform to hierarchical manifolds. This is why some previous works focus on learning or embedding hierarchical structures within the latent space [1, 2]. Furthermore, hierarchical relationships in real-world data may not always be represented by nested structures. As such, this assumption appears somewhat oversimplified. If the intention was to convey that the data does not inherently lie on a hierarchical manifold but must be trained to conform to one, it would be beneficial to clarify this in the writing.

3. Since the hypothesis regarding hierarchical manifolds is not clear, it is difficult to understand how graph-diffusion inference can be leveraged even in the absence of HMCF (Figure 1(a)). In other words, what is the relationship between graph-diffusion inference and hierarchical manifold? Isn’t it possible to conduct graph diffusion-based inference using the relationships among all labels, even without the discussion about the hierarchical manifold?

4. Some sections of the writing appear unorganized. For example, in Line 84, the abbreviation "HMCF" is introduced before it is defined. Additionally, in Line 188,Is “14,036” referring to the number of test samples or the total number of hierarchical labels? Making these points clearer would enhance the conciseness of the writing.



**Originality**

5. The hierarchical contrastive learning method that utilizes labels at all levels has been proposed previously [3], so its novelty here is somewhat limited. It would be helpful to clarify how the proposed approach differs from [3].

**Quality**

6. Relatedly, there are many additional experiments in the Appendix that are not referenced in the main text (except for Table 5). If these experiments are relevant and essential to this method, it would be helpful to mention them in the main text so that readers can find them.

7. Some important works seem to be missing from the Related Work section. In addition to top-down and bottom-up approaches, there are hierarchical classification studies that use multi-branch architectures to predict all levels simultaneously based on the complete label hierarchy [4, 5]. A comparison with these approaches would be helpful. These studies aim to address the question raised in Lines 163–164: “Can the predictions across different levels in the category hierarchy mutually reinforce each other to improve overall accuracy?” and are driven by similar motivations.


8. As a minor point, citations would be easier to read if placed in parentheses.

[1] HIER: Metric Learning Beyond Class Labels via Hierarchical Regularization, (CVPR, 2023)
[2] Learning Structured Representations by Embedding Class Hierarchy, (ICLR, 2023)
[3] Use All The Labels: A Hierarchical Multi-Label Contrastive Learning Framework (CVPR, 2022)
[4] Your “Flamingo” is My “Bird”: Fine-Grained, or Not (CVPR, 2021)
[5] Label Relation Graphs Enhanced Hierarchical Residual Network for Hierarchical Multi-Granularity Classification (CVPR, 2022)

**Questions:**

1. Please refer to the Weaknesses:

The most important questions are regarding the hypothesis (W1-3), missing related work (W7), and the originality (W5).

2. Here are additional questions:

1) Isn’t the inference cost for graph-diffusion inference much higher compared to bottom-up or top-down approaches?

2) Which metric is the most important in the evaluation (L317-323)?

3) Could incorrect predictions also propagate through graph diffusion and negatively impact inference?

---

### Note · Authors · 2024-11-25

I have read and agree with the venue's withdrawal policy on behalf of myself and my co-authors.